# Research on Adaptive Multi-Source Information Fault-Tolerant Navigation Method Based on No-Reference System Diagnosis

**DOI:** 10.3390/s19132911

**Published:** 2019-07-01

**Authors:** Ling Zhang, Yuchen Cui, Zhi Xiong, Jianye Liu, Jizhou Lai, Pin Lv

**Affiliations:** 1College of Automation Engineering, Nanjing University of Aeronautics & Astronautics, Nanjing 210016, China; 2Jiangsu Key Laboratory of Internet of Things and Control Technologies, Nanjing 211106, China

**Keywords:** MAV, no-reference system, fault detection, fault tolerance

## Abstract

In order to obtain accurate and optimized navigation sensor information, it is necessary to study information fusion and fault diagnosis with high reliability, high precision and high autonomy, and then to propose a rapid and accurate intelligent decision-making scheme based on multi-source and heterogeneous navigation information. In view of the existing fault-tolerant navigation federated filter structure, the method of assuming the reference system (inertial navigation system) to be fault-free and then diagnosing the measuring sensor fault is generally adopted. Considering that the structure of the filter can’t detect and isolate the faults of the inertial navigation system, the performance of the MEMS inertial navigation system declines due to complex environments resulting from vibrations and temperature changes; additionally, external interference may lead to the direct failure of the MEMS inertial device. Therefore, this paper studies a fault-tolerant navigation method based on a no-reference system. For the sensor sub-system of a custom micro air vehicle (MAV), a fault detection method based on a reference-free system is proposed. Based on the fault type analysis, some improvements have been made to the existing residual chi-square detection method, and an interactive residual fault detection method with distributed states is proposed. On this basis, aiming at the characteristics of a reference-free system, the weight distribution scheme of the reference system and the tested systems are studied, and a self-regulation filter fusion and fault detection method based on reference-free system is designed.

## 1. Introduction

MAVs (Micro Air Vehicles) are characterized by small size, light weight, low cost, strong flexibility and good concealment. In recent years, with the gradual maturity of the relevant technology, MAVs have been widely used in military and civil fields. For example, they have been used in battlefield reconnaissance, surveillance, target search and positioning. In civil fields, they have been used in agriculture and forestry, aerial photography and electric power [1]. MAVs have broad application prospects in the military and civil fields, with many countries showing interest in the research of relevant technologies [2,3,4,5,6,7].

The complex flying environment of MAVs has brought about challenges regarding the reliability of navigation systems. Due to the limitations of cost, volume, load and power consumption, micro and small aircraft usually use MEMS (Micro-Electro-Mechanical System) navigation sensors. Compared with the high-precision navigation sensors (such as laser/ fiber-optic gyroscope) commonly used in large aircraft, the accuracy and reliability of MEMS navigation sensors are poor. The performance of the MEMS navigation sensor will decrease or even fail due to the complex flying environment of MAVs (such as vibration, temperature change, wind speed change, electromagnetic interference, etc.).

In order to improve the reliability of the navigation system of MAVs, fault-tolerant navigation ability should be improved. Fault-tolerant navigation systems can maintain stability and meet certain performance indexes in case of sensor failure. The implementation of fault-tolerant navigation generally relies on the system redundancy configuration. The navigation system can be reconstructed by the redundancy configuration of the navigation sensor and the replacement of fault parts. Compared with hardware redundancy, analytical redundancy has the advantages of low cost, light weight and small size, and is suitable for MAVs. In recent years, aircraft navigation systems have shown the trend of diversification and technology maturity, which promotes the development of the analytical redundancy fault-tolerant navigation technology. A Fault Tolerant SINS/GPS/CNS Integrated Navigation Scheme was proposed by the team of Fang Jiancheng of the Beijing University of Aeronautics and Astronautics [8]. It adopts the framework of federal filter, and takes the inertial navigation system as the reference to diagnose satellite navigation system and astronomical navigation system faults. A fault tolerant navigation system based on GPS/INS/vision sensors scheme is proposed by the University of New South Wales in Australia for unmanned aerial vehicles [9]. Beijing Jiaotong University proposed a fault-tolerant navigation system based on the INS/GPS/Locata scheme [10]. Locata is a regional radio navigation system. When the satellite navigation system is limited in some areas, the navigation system can be reconstructed through Locata. In this paper, a fault-tolerant navigation method based on the no-reference system is studied [11,12,13,14,15,16]. For sensors of a custom MAV, a fault detection method based on reference-free system is proposed. Based on the analysis of fault type, some improvements have been made to the existing residual chi-square detection method, and an interactive distributed residual fault detection method is proposed. On this basis, aiming at the characteristics of the no-reference system, the weight distribution scheme of the reference system and the information distribution coefficient is studied, the self-regulation filter fusion method based on no-reference fault detection is designed, and the simulation experiment is carried out through the digital simulation platform, which verifies the reliability of the method.

## 2. Analysis of Multi-Type Sensor Fault Model 

The output of measurement information of each sub-system navigation sensor in the integrated navigation system determines the precision and reliability of the integrated navigation system. When each navigation sensor is not in its normal working environment or the sensor fails, it will output wrong navigation information, thus affecting the navigation performance of the navigation system. As we all know, there are many reasons that lead to the failure of different types of navigation sensors on the high-altitude reconnaissance drone, but by summarizing and concluding the output fault information of sensor, the faults can be basically divided into two categories: hard and soft faults. Hard faults are caused by sudden changes of measurement information, while soft faults are caused by slow changes of measurement information over time. In my opinion, the difference between the two is that hard faults are sudden faults and soft faults are slow fault. Hard faults are time-independent, while soft faults are time-dependent.

(1) Hard Fault

Hard faults can be represented as shown in Figure 1.

In the figure above, the abscissa is time and the ordinate is the fault value. At time $t_{0}$, a hard fault with value of count occurs. The expression form of the above figure is:(1)$$y=\{\begin{array}{c} count,(t>t_{0})\\ 0,\left(t\leq t_{0}\right) \end{array}$$
where t is the time, y is the fault value, and count is the size of the fault information when the fault occurs.

(2) Soft Fault

Soft faults can be represented as shown in Figure 2 and Figure 3.

Figure 2 shows the linear soft fault, while Figure 3 shows the quadratic fault. As shown in the figure, the y-coordinate is the fault value, the x-coordinate is time, and the fault in the figure above can be expressed as:(2)$$y=\{\;\begin{array}{c} k\left(t-t_{0}\right),\mathrm{linear}\;\mathrm{soft}\;\mathrm{fault}\\ \;k{\left(t-t_{0}\right)}^{2},\mathrm{quadratic}\;\mathrm{soft}\;\mathrm{fault} \end{array}$$
where y is the fault size, k is the fault coefficient, $t_{0}$ is the start time of the fault, and t is the system running time.

To sum up, the sudden fault of step information is very unfavorable to the system, and some systems may even be damaged due to the fault of large abrupt changes. Meanwhile, since k is related to the time-varying condition, it is difficult for the system to detect the existence of soft faults if the measured fault information changes slowly over time, which puts forward strenuous requirements for the fault detection algorithm.

## 3. Research on Interactive Fault Detection Method Based on No-Reference System

### 3.1. Analysis of Residual Chi-Square Fault Detection Method

Residual $\chi^{2}$ test is a system-level fault detection algorithm which can judge whether a set of measurement information in the kalman filter system is effective in real time. However, if a fault is detected, the specific cause of the fault can’t be determined. In this section, the basic principle of the residual $\chi^{2}$ test method is given [17,18].

In the federated filter of multi-information fusion, for each sub-kalman filter, its residual value can be expressed as:(3)$$\gamma_{i}\left(k\right)=Z_{i}\left(k\right)-H_{i}\left(k\right){\hat{X}}_{i}\left(k/k-1\right)$$

The one-step predicted value ${\hat{X}}_{i}\left(k/k-1\right)$ in the above equation is:(4)$${\hat{X}}_{i}\left(k/k-1\right)=\Phi_{i}\left(k/k-1\right){\hat{X}}_{i}\left(k-1\right)$$
where $\gamma_{i}$ is residual value, $Z_{i}$ is observation, $H_{i}$ is observation coefficient, ${\hat{X}}_{i}$ is the one-step predicted value, and $\Phi_{i}$ is predicted state coefficient.

When a sub-system is trouble-free, the residual $\gamma_{i}\left(k\right)$ of corresponding sub-filter is Gaussian white noise with zero mean value. Therefore, the variance of sub-filter can be expressed as:(5)$$A_{i}\left(k\right)=H_{i}\left(k\right)P_{i}\left(\frac{k}{k}-1\right)H_{i}^{T}\left(k\right)+R_{i}\left(k\right)$$

According to the feature of whether the mean of the residuals $\gamma_{i}\left(k\right)$ of the sub-filters corresponding to the navigation sub-system is zero, the fault of the sub-navigation system is determined.

Suppose the system fails    $E\left\{r_{i}\left(k\right)\right\}=\mu ,E\left\{\left[r_{i}\left(k\right)-\mu \right]{\left[r_{i}\left(k\right)-\mu \right]}^{T}\right\}=A_{k}$

The system is normal     $E\left\{r_{i}\left(k\right)\right\}=0,E\left\{r_{i}\left(k\right)r_{i}^{T}\left(k\right)\right\}=A_{k}$

The fault detection function of residuals $\chi^{2}$ is:(6)$$\lambda_{i}\left(k\right)=\gamma_{i}^{T}\left(k\right)A_{i}^{-1}\left(k\right)\gamma_{i}\left(k\right)$$
where $\lambda_{i}\left(k\right)$ represents a $\chi^{2}$ distribution function which is subject to the degree of freedom m (m represents the dimension of the observation matrix), then $\lambda_{i}\left(k\right)\sim \chi^{2}$. When the false alarm rate $P_{f}$ is set, the corresponding fault threshold value $T_{D}$ can be obtained by looking up the table. Therefore, the fault judgment standard can be expressed as:(7)$$\begin{array}{c} \lambda_{i}\left(k\right)>T_{D}\;\;\;\;\;\mathrm{the}\;\mathrm{system}\;\mathrm{fails}\\ \lambda_{i}\left(k\right)<T_{D}\;\mathrm{the}\;\mathrm{system}\;\mathrm{is}\;\mathrm{normal} \end{array}\;\}$$

$\chi^{2}$ fault detection algorithm is one of the commonly used fault detection algorithms at present, which is mainly divided into state $\chi^{2}$ inspection method and residual $\chi^{2}$ inspection method. Because of its simple algorithm principle and small amount of calculation, the residual $\chi^{2}$ test method can detect the mutation information in the measurement information very well. However, the residual $\chi^{2}$ inspection method has a weak ability to detect the slowly changing fault information with time. Therefore, based on the residual $\chi^{2}$ inspection method, this chapter improves and perfects it to enable its improved algorithm to detect both hard and soft faults of the system at the same time, improve the system’s ability to detect faults, and make the integrated navigation system more reliable and stable.

### 3.2. Research on Interactive Dispersion Residual $\chi^{2}$ Fault Detection Algorithm

In the last section analysis of chi-square fault detection method based on residual, according to the proposed fault detection scheme based on no-reference model, refer to interaction pattern of multiple model algorithm. The six sub-filter models consisting of INS (inertial navigation system)/Optical Flow, INS/Lidar, INS/Virtual INS, Optical Flow/Virtual INS, Lidar/Virtual INS and Optical Flow/Lidar are analyzed, and the residual fault $\chi^{2}$ detection algorithm of each sub-filter model is studied.

(1) Interactive Input of Various Types of Sensor Solution Information

The equation of state of the multi-information fusion navigation system is:(8)$$\dot{X}\left(t\right)=A\left(t\right)X\left(t\right)+G\left(t\right)W\left(t\right)$$

The state vector X is:(9)$$X=[\begin{array}{ccccccccc} \varphi_{E} & \varphi_{N} & \varphi_{U} & \delta V_{E} & \delta V_{N} & \delta V_{U} & \delta L & \delta \lambda & \delta h{]}^{T} \end{array}$$
where $\varphi_{E}$, $\varphi_{N}$,$\;\varphi_{U}$ are the error Angle of the platform, $\delta V_{E}$,$\;\;\delta V_{N}$, $\delta V_{U}$ the speed error of the northeast celestial direction, δL, δλ, δh latitude, longitude and height position error. A(t) is the state coefficient matrix; G(t) is the error coefficient matrix; W(t) is the random error vector of white noise.

The measurement equation of each sub-filter is analyzed as follows:➢Velocity measurement equation of SINS/Optical Flow
(10)$$Z_{k}=\left[\begin{array}{c} v_{e,I}-v_{e,p}\\ v_{n,I}-v_{n,p} \end{array}\right]=h\left(t\right)X\left(t\right)+n$$
where $v_{e,I}$,$\;v_{n,I}$ are the velocity output calculated by the inertial navigation system under the navigation coordinate system, and $v_{e,p}$,$\;v_{n,p}$ are the velocity output calculated by the optical flow sensor under the navigation coordinate system.➢Position and velocity measurement equations of SINS/ Lidar
(11)$$Z_{k}=\left[\begin{array}{c} r_{e,I}-r_{e,L}\\ r_{n,I}-r_{n,L} \end{array}\right]=H\left(t\right)X\left(t\right)+n$$
(12)$$Z_{k}=\left[\begin{array}{c} v_{e,I}-v_{e,L}\\ v_{n,I}-v_{n,L} \end{array}\right]=H\left(t\right)X\left(t\right)+n$$
where $r_{e,I}$,$\;r_{n,I}$ are the position output calculated by the inertial navigation system under the navigation coordinate system;$\;r_{e,L}$,$\;r_{n,L}$ are the position output calculated by the Lidar sensor under the navigation coordinate system;$\;v_{e,I}$,$\;v_{n,I}$ are the velocity output calculated by the inertial navigation system under the navigation coordinate system;$\;v_{e,L}$,$\;v_{n,L}$ are the velocity output calculated by the Lidar sensor under the navigation coordinate system.➢Velocity measurement equation of Optical Flow/ Lidar
(13)$$Z_{k}=\left[\begin{array}{c} v_{e,L}-v_{e,p}\\ v_{n,L}-v_{n,p} \end{array}\right]=h\left(t\right)X\left(t\right)+n$$
where $v_{e,L}$,$\;v_{n,L}$ are the velocity output calculated by the inertial navigation system under the navigation coordinate system, and $v_{e,p}$,$\;v_{n,p}$ are the velocity output calculated by the optical flow sensor under the navigation coordinate system.➢Velocity measurement equation of Optical Flow/ Virtual INS
(14)$$Z_{k}=\left[\begin{array}{c} v_{e,VI}-v_{e,p}\\ v_{n,VI}-v_{n,p} \end{array}\right]=h\left(t\right)X\left(t\right)+n$$
where $v_{e,VI}$,$\;v_{n,VI}$ are the velocity output calculated by the inertial navigation system under the navigation coordinate system, and $v_{e,p}$,$\;v_{n,p}$ are the velocity output calculated by the optical flow sensor under the navigation coordinate system.➢Position and velocity measurement equations of Lidar/ Virtual INS
(15)$$Z_{k}=\left[\begin{array}{c} r_{e,VI}-r_{e,L}\\ r_{n,VI}-r_{n,L} \end{array}\right]=H\left(t\right)X\left(t\right)+n$$
(16)$$Z_{k}=\left[\begin{array}{c} v_{e,VI}-v_{e,L}\\ v_{n,VI}-v_{n,L} \end{array}\right]=H\left(t\right)X\left(t\right)+n$$
where $r_{e,VI}$,$\;r_{n,VI}$ are the position output calculated by the inertial navigation system under the navigation coordinate system;$\;r_{e,L}$,$\;r_{n,L}$ are the position output calculated by the Lidar sensor under the navigation coordinate system;$\;v_{e,VI}$,$\;v_{n,VI}$ are the velocity output calculated by the inertial navigation system under the navigation coordinate system, and $v_{e,L}$,$\;v_{n,L}$ are the velocity output calculated by the optical flow sensor under the navigation coordinate system.➢Position and attitude measurement equations of Laser SINS/Virtual INS
(17)$$Z_{k}=\left[\begin{array}{c} r_{e,L}-r_{e,VI}\\ \begin{array}{c} r_{n,L}-r_{n,VI}\\ r_{u,L}-r_{u,VI} \end{array} \end{array}\right]=H\left(t\right)X\left(t\right)+n$$
(18)$$Z_{k}=\left[\begin{array}{c} \gamma_{I}-\gamma_{VI}\\ \begin{array}{c} \theta_{I}-\theta_{VI}\\ \varphi_{I}-\varphi_{VI} \end{array} \end{array}\right]=H\left(t\right)X\left(t\right)+n$$
where $r_{e,I}$,$\;r_{n,I}$ are the position output calculated by the inertial navigation system under the navigation coordinate system;$\;r_{e,VI}$,$\;r_{n,VI}$ are the position output calculated by the virtual inertial navigation system under the navigation coordinate system; $\gamma_{I}$, $\theta_{I}$,$\;\varphi_{I}$ are the attitude output calculated by the inertial navigation system under the navigation coordinate system, and $\gamma_{VI}$ , $\theta_{VI}$, $\varphi_{VI}$ are the attitude output calculated by the virtual inertial navigation system under the navigation coordinate system.

(2) Establish the fault detection function based on sliding mode predictor

Slow change fault will lead to slow growth of error of navigation sub-system. Before being judged as fault, this sub-system is still normally connected to the main filter, and is isolated instantaneously until the error increases to the critical value. The delayed processing of fault makes the system contaminated by fault information. In view of this, the idea that the n-time one-step state prediction estimation does not contain fault information, but that measurement accumulation can amplify the fault, is considered. So, some improvements to the traditional chi-square detection method have been achieved.

First, consider the current situation at time *k*. After *n* times of prediction from (*k-nt*) to moment *k*, the current state prediction ${\hat{X}}_{k|k-1}$ at moment k can be obtained as follows:(19)$${\hat{X}}_{k|k-1}=\overset{k-1}{\underset{k+\left(j-n\right)T-1}{\prod }}\Phi_{\left(k+\left(j-n\right)T\right)/\left(k+\left(j-n\right)T-1\right)}{\hat{X}}_{\left(k+\left(j-n\right)T\right)}$$

In the above formula, ${\hat{X}}_{\left(k+\left(j-n\right)T\right)}$ does not contain fault information. Further, the predicted measured value at time *nT* was calculated as:(20)$${\hat{Z}}_{k}=H_{k}{\hat{X}}_{k|k-1}$$

According to the formula, there is no fault information in ${\hat{Z}}_{k}$.

At this point, $P_{k}$ is calculated by the following formula:(21)$$P_{k|k-1}=\overset{k-1}{\underset{k+\left(j-n\right)T-1}{\prod }}(\Phi_{\left(k+\left(j-n\right)T\right)/\left(k+\left(j-n\right)T-1\right)}P_{\left(k+\left(j-n\right)T\right)}\Phi_{{\left(k+\left(j-n\right)T\right)/\left(k+\left(j-n\right)T-1\right)}^{T}}+Q_{\left(k+\left(j-n\right)T\right)})$$
where the setting and selection of the recursive time period *nT* are related to system performance.

Thus, the residual error and its variance are:(22)$$r_{k}=Z_{k}-{\hat{Z}}_{k}=Z_{k}-H_{k}{\hat{X}}_{k|k-1}$$

(23)$$A_{k}=H_{k}P_{k|k-1}{H_{k}}^{T}+R_{k}$$

Further, the fault detection function can be established as follows:(24)$$\lambda_{k}={r_{k}}^{T}{A_{k}}^{-1}r_{k}$$

Similarly, the fault criteria are:(25)$$J_{i}\left(k\right)=\{\begin{array}{c} 1\;\;\;\lambda_{k}>T_{D}\\ 0\;\;\lambda_{k}\leq T_{D} \end{array}$$

$T_{D}$ is a preset threshold. For those sub-filters such as SINS/Lidar and SINS/Optical Flow, here take i = 1, 2, ..., 6.

(3) Calculate the fault detection result at time *k* and locate the fault

First, the logical relation based on inertial navigation system is established as:(26)$$F_{SINS}\left(k\right)=J_{1}\left(k\right)\wedge J_{2}\left(k\right)\wedge J_{6}\left(k\right)$$

When $F_{SINS}\left(k\right)=1$, the inertial navigation system fails; when $F_{SINS}\left(k\right)=0$, the inertial navigation system does not fail. Where, $J_{1}\left(k\right)$, $J_{2}\left(k\right)$ and $J_{6}\left(k\right)$ respectively represent the fault detection values of three sub-systems: SINS/Optical Flow, SINS/ Lidar and SINS/Virtual SINS.

Secondly, the logical relation based on Lidar is established:(27)$$F_{L}\left(k\right)=J_{2}\left(k\right)\wedge J_{3}\left(k\right)\wedge J_{5}\left(k\right)$$

When $F_{L}\left(k\right)=1$, the Lidar fails; when $F_{L}\left(k\right)=0$, the Lidar does not fail. Where, $J_{2}\left(k\right),$
$J_{3}\left(k\right)$ and $J_{5}\left(k\right)$ respectively represent the fault detection values of three sub-systems: SINS/Lidar, Optical Flow/Lidar and Lidar/Virtual SINS.

Thirdly, the logical relation based on Optical Flow is established:(28)$$F_{P}\left(k\right)=J_{1}\left(k\right)\wedge J_{3}\left(k\right)\wedge J_{4}\left(k\right)$$

When $F_{P}\left(k\right)=1$, the Optical Flow fails; when $F_{P}\left(k\right)=0$, the Optical Flow does not fail. Where,$\;J_{1}\left(k\right),$
$J_{3}\left(k\right)$ and $J_{4}\left(k\right)$ respectively represent the fault detection values of three sub-systems: SINS/Optical Flow, Optical Flow/Lidar and Optical Flow/Virtual SINS.

Finally, the logical relation based on the Virtual SINS is established:(29)$$F_{V}\left(k\right)=J_{4}\left(k\right)\wedge J_{5}\left(k\right)\wedge J_{6}\left(k\right)$$

When $F_{V}\left(k\right)=1$, the Virtual SINS fails; when $F_{V}\left(k\right)=0$, the Virtual SINS does not fail. Where, $J_{4}\left(k\right),$
$J_{5}\left(k\right)$ and $J_{6}\left(k\right)$ respectively represent the fault detection values of three sub-systems: Optical Flow/Virtual SINS, Lidar/Virtual SINS and SINS/Virtual SINS.

### 3.3. Research on Self-Regulation Filter Fusion Method of No-Reference Fault Detection

Combined with the no-reference mode of fault detection, an improved self-tuning filtering fusion method is proposed. Considering the traditional federated filtering algorithm, the inertial navigation system is selected as the public reference system, and the output of the inertial navigation system is directly sent to the main filter on the one hand, and to each sub-filter as the measured value on the other. The no-reference fault detection proposed in this paper considers that the inertial navigation system may also fail, and gives detection and judgment thereof. Therefore, the traditional filtering architecture is improved with the federated filter structure. Firstly, the reference system is selected, and according to the fault detection results of the inertial navigation system and virtual inertial navigation system, the inertial navigation system is selected first to establish the equation of state, and the sub-filter is established by combining the measurement equation of the corresponding sensor sub-system. Secondly, the corresponding fault detection values are obtained through the no-reference fault detection method to judge the fault condition of each sub-sensor, and the new information distribution coefficient of the filter is determined by the fault detection values. Finally, based on the fault judgment results, the fault-free sub-filter is selected, and the state recursive results during fault detection are combined to complete the further federated filtering algorithm to achieve the optimal estimation of the multi-source navigation system.

The state equation and measurement equation of the established system are as follows:(30)X(k)=Φ(k|k−1)H(k−1)+Γ(k−1)W(k−1)
(31)Z(k)=H(k)X(k)+V(k)
where $\Phi_{k|k-1}$ refers to the transfer matrix corresponding to the equation of state of the system, and $W_{k-1}$ is the white noise random error vector corresponding to the error equation of the inertial navigation system.

Each sub-system constitutes a sub-filter, and the corresponding state equation and measurement equation are established as follows:(32)$$X_{i}\left(k\right)=\Phi_{i}\left(k|k-1\right)X_{i}\left(k-1\right)+\Gamma \left(k-1\right)W_{i}\left(k-1\right)$$

(33)$$Z_{i}\left(k\right)=H_{i}\left(k\right)X_{i}\left(k\right)+V_{i}\left(k\right)$$

Here, X(k) , $X_{i}\left(k\right)$ is the same state quantity, then the global optimal estimation can be expressed as:(34)$$P_{g}={\left(\overset{N}{\underset{i=1}{\sum }}P_{i}^{-1}\right)}^{-1}$$

(35)$${\hat{X}}_{i}=P_{g}\left(\overset{N}{\underset{i=1}{\sum }}P_{i}^{-1}{\hat{X}}_{i}\right)$$

Thus, the information distribution of each sub-filter is:(36)$$P_{i}=\beta_{i}^{-1}P_{g}$$
(37)$${\hat{X}}_{i}={\hat{X}}_{g}$$
where ${\hat{X}}_{g}$ is the global optimal estimation, $P_{g}$ is the covariance matrix of the state variable, ${\hat{X}}_{i}$ is the optimal estimation of the sub-filter, $P_{i}$ is the covariance matrix of the state variable of the corresponding sub-filter, and $\sum_{i=1}^{N}\beta_{i}=1$.

In the design of federated filter, the determination of information allocation coefficient is very important. $\beta_{i}$ is defined as the "information allocation coefficient" of the information distribution obtained by the main filter and each sub-filter, and satisfies the principle of information conservation:(38)$$\overset{N}{\underset{i=1}{\sum }}\beta_{i}=1,0<\beta_{i}<1$$

Considering that the variance of the residual in kalman filter reflects whether the sub-system is faulty, the information allocation coefficient can be adjusted according to the relationship between the residual variance and the fault detection threshold, and the expression of the information allocation coefficient can be established as following:(39)$$\beta_{i}=\frac{{\left(T_{Di}/\lambda_{i}\left(k\right)\right)}^{2}}{\sum_{i=1}^{N}{\left(T_{Di}/\lambda_{i}\left(k\right)\right)}^{2}}$$

Substitute into Equation (35) to calculate the corresponding covariance matrix. Where $\lambda_{i}\left(k\right)$ is the residual chi-square fault detection function value of the *i-*th sub-system, and $T_{Di}$ is the fault threshold value set by the *i-*th sub-system. From $\lambda_{i}\left(k\right)$, we can know the fault condition of the sub-system. When $T_{Di}/\lambda_{i}\left(k\right)>1$, the sub-system does not fail; when $0<T_{Di}/\lambda_{i}\left(k\right)<1$, the sub-system fails. Therefore, the information allocation coefficient formula, as shown in Equation (39), can be used for dynamic adjustment according to the fault situation of the sub-system, so as to get better fusion results.

On this basis, a multi-information fusion scheme based on no-reference system fault detection is proposed, as shown in Figure 4. First, according to the fault detection results, the inertial navigation system is preferred to establish the state equation, and the sub-filter is established by combining the measurement equation of the corresponding sensor sub-system. Second, the corresponding fault detection values are obtained through the no-reference fault detection method to judge the fault situation of each sub-sensor. Third, based on the fault judgment results, the corresponding sub-filter is selected and the state recursive results during fault detection are combined to carry out the next federal filtering algorithm. Fourth, the corresponding fault detection values are obtained through the no-reference fault detection method to judge the fault situation of each sub-sensor. Finally, based on the fault judgment results, the corresponding sub-filter is selected and the state recursive results during fault detection are combined to carry out the next federal filter algorithm.

## 4. Verification and Analysis of Self-Regulation Fusion Method Based on Fault-Free Detection 

### 4.1. Self-Regulation Fusion Simulation Condition Setting

Combined with the research and analysis of the fault detection of the no-reference system and the corresponding self-regulation fusion method in this chapter, the simulation verification analysis will be given in this section. First, the parameters of the airborne sensors of the MAV are set, and the parameters of each sensor are shown in Table 1 and Table 2 respectively. Second, the output update frequency of inertial navigation system is 50 Hz, the output update frequency of Lidar is 10 Hz, and the output update frequency of Optical Flow sensor is 2 Hz. In order to analyze the necessity and effectiveness of the self-regulation fusion method based on the no-reference system, the fault information of inertial devices and Optical Flow sensors is set in this section.

Secondly, the fault types and sizes of inertial devices and optical flow sensors are set. In this section, both types of sensor faults are set to a hard fault type. Among them, the fault size of the gyro is set at 4°/s in the period between 120–150 s, the fault size of the accelerometer is set at 4 m/s, and the fault size of the optical flow sensor is set at 6 m/s in the period between 900–940 s.

### 4.2. Analysis of Simulation Results of Self-Regulation Fusion

In order to verify the self-regulation fusion method based on no-reference fault detection and analyze and explain the influence of state and measurement faults on the navigation performance of the system, this section respectively verifies the navigation performance of the system from the inertial device fault and optical flow fault. In addition, performance analysis is also carried out for the unknown state fault, and the necessity of no-reference detection is analyzed from the perspective of system reliability.

(1) In the case of fault of measurement information

The results of self-adjustment fusion in the case of fault of measurement information are shown in Figure 5. The blue dotted line ‘1’ in the figure represents the navigation results in the case of fault-free of the system, and the red solid line ‘2’ represents the navigation results in the case of fault of optical flow sensor. Among them, the fault occurs between 900 and 940 s.

As can be seen from the above simulation results, after the optical flow sensor fails and isolates in the period between 900–940 s, the navigation result of the system is affected to a certain extent, and the error curve changes to different degrees. However, as can be seen from the position and velocity error curves of the system, the error fluctuates in the same order of magnitude. The comparative data analysis of position errors of the optical flow sensor with and without faults is shown in Table 3. It can be seen that there is no significant difference in the size of position error between the two cases. This indicates that the self-regulation fusion method can guarantee the navigation accuracy of the system for a period of time after the sensor fault.

(2) In the case of fault of Status information 

The results of self-regulation fusion in the case of fault of state information are shown in Figure 6. The blue dotted line in the figure represents the navigation results in the case of fault-free of the system, and the red solid line represents the navigation results in the case of fault of the inertial sensor. Among them, the fault occurred in the period between 120–150 s; virtual inertial navigation is used to replace the airborne inertial navigation system after the fault is determined.

As can be seen from the above simulation results, after the inertial sensor fails and is isolated in the period between 120–150 s, the virtual inertial navigation system is selected to replace it. In the simulation, it is assumed that the accuracy of the virtual inertial device is one order of magnitude lower than that of the airborne inertial device, and the velocity error curve has different degree of mutation, so that the time period of the fault can be calculated. However, as can be seen from the position error curve of the system, the error still fluctuates in the same order of magnitude. As shown in Table 4, the comparative data analysis of the position error of the inertial sensor with and without faults shows that there is no obvious difference in the size of the position error between the two cases. Similar to the result analysis of the measured information fault above, the self-regulation fusion method can guarantee the navigation accuracy of the system within a period of time after the sensor fault occurs.

(3) In cases where state information fails but is not isolated

In order to illustrate the necessity of fault detection based on the no-reference system, this section assumes the fault of the inertial sensor, and carries out simulation analysis on the three cases of fault-free, fault judgment and isolation, and fault undetermined. The comparison of fusion results is shown in Figure 7. The blue dotted line ‘1’ in the figure represents the navigation results in the case of fault-free of the system, the red solid line ‘2’ represents the self-regulation fusion navigation results in the case of fault of the inertial sensor, and the green dotted line ‘3’ indicates the navigation result of the system when the inertial sensor fails but is not isolated.

As can be seen from the above simulation results, due to the fault of the inertial sensor without isolation, the airborne navigation information is contaminated, the accuracy is reduced by two orders of magnitude, and the system performance is greatly affected. Therefore, based on a fault-tolerant navigation architecture that assumes a fault-free reference system (inertial navigation system), because of the condition that the inertial navigation system is taken as a reference system and fails, the fault of the measurement sensor is judged, this method cannot realize fault detection and fault-tolerant navigation of all navigation sensors in the system. It also shows the necessity of the self-regulation fusion method based on no-reference fault detection, which ensures the accuracy of the system.

## 5. Conclusions

Considering that the MEMS inertial navigation system of MAV is not completely reliable, the federated filter structure commonly used in fault-tolerant navigation assumes that the inertial navigation system is fault-free. In this paper, a fault detection method based on the no-reference system is proposed, which successively detects the faults of multiple sub-sensor systems contained in MAV, and uses the interactive residual detection method of dispersed state to realize the overall fault diagnosis of sub-sensor systems. On this basis, an improved self-regulation filter fusion method is proposed, which eliminates the assumption that the inertial navigation system is fault-free and determines the common reference system based on the no-reference fault detection results. According to the fault detection value, the information distribution coefficient of the filter is adjusted in real time to realize the intelligent and autonomous fusion of multi-source navigation information and improve the reliability and stability of the navigation system under fault conditions.

## Figures and Tables

**Figure 1 sensors-19-02911-f001:**
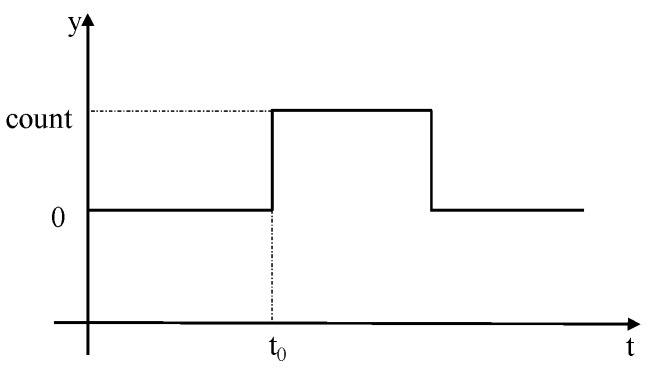
Schematic Diagram of Hard Fault Information.

**Figure 2 sensors-19-02911-f002:**
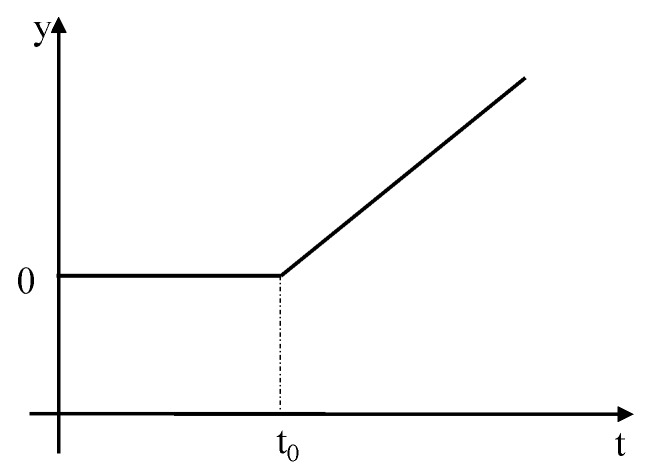
Schematic Diagram of Linear Soft Fault.

**Figure 3 sensors-19-02911-f003:**
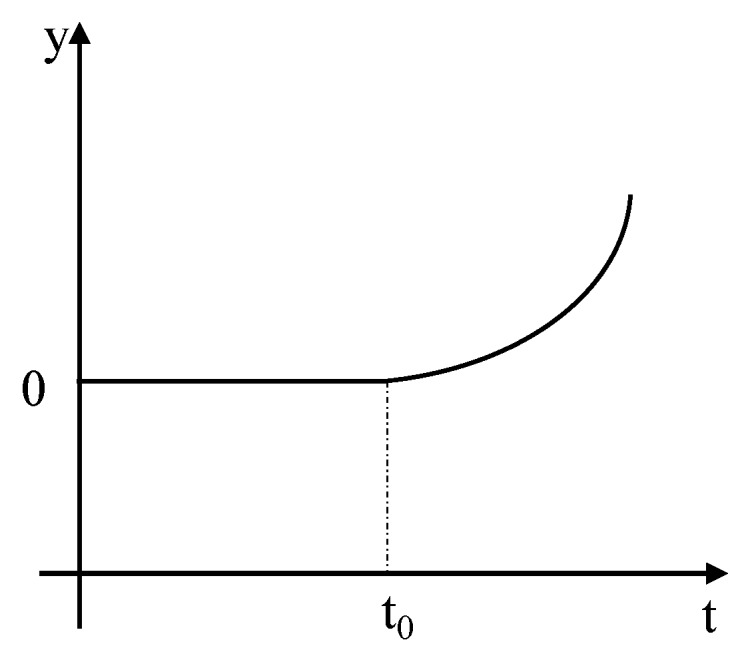
Schematic Diagram of Quadratic Soft Fault.

**Figure 4 sensors-19-02911-f004:**
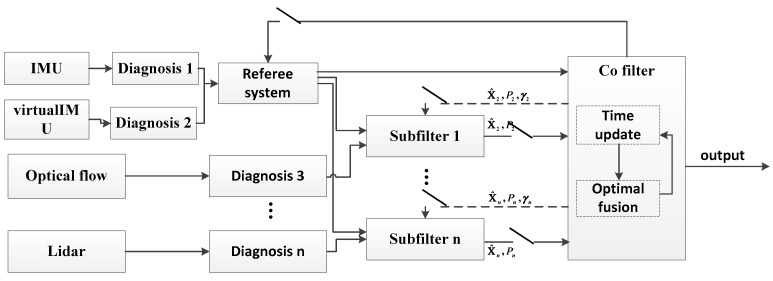
Information Fusion Scheme Diagram Based on No-reference System Fault Detection.

**Figure 5 sensors-19-02911-f005:**
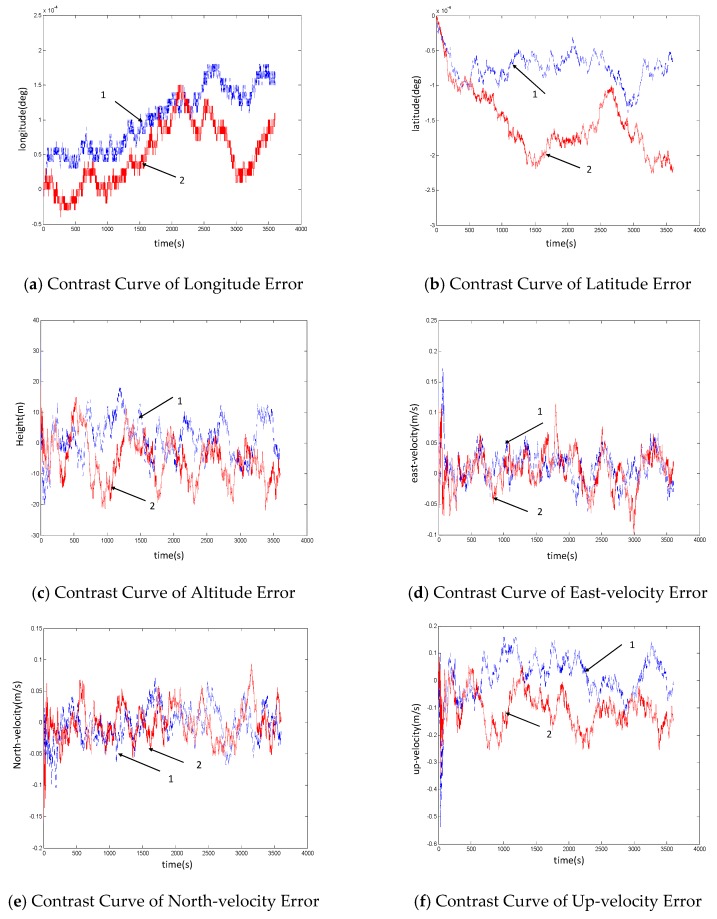
Contrast Diagram of System Self-regulation Fusion Navigation Results When the Optical Flow Sensor Fails.

**Figure 6 sensors-19-02911-f006:**
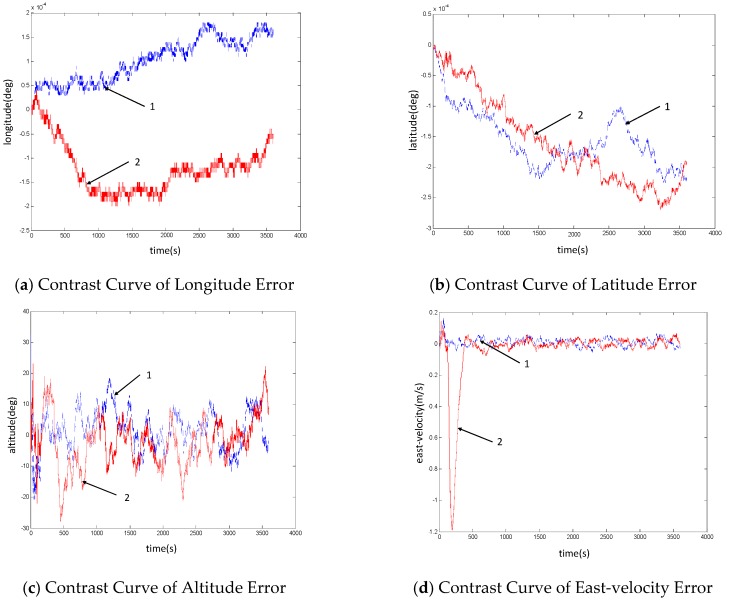
Contrast Diagram of System Self-regulation Fusion Navigation Results When the inertial sensor Fails.

**Figure 7 sensors-19-02911-f007:**
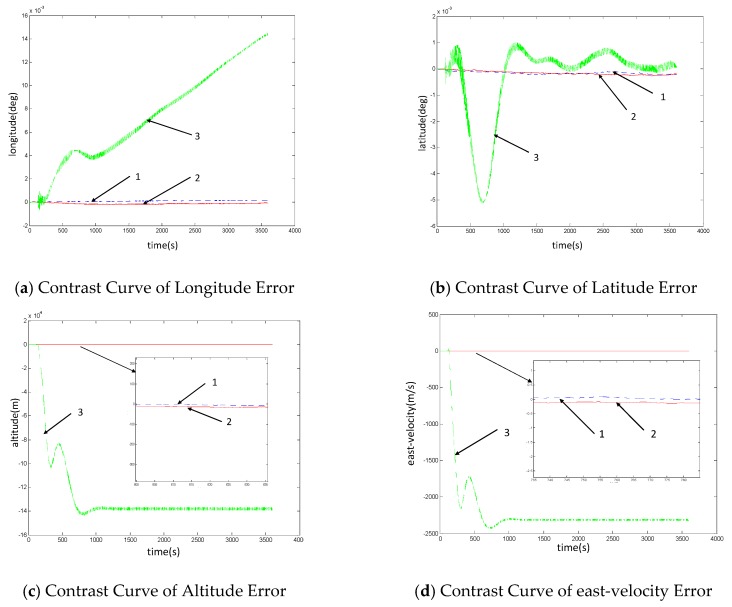
Contrast Diagram of System Fusion Navigation results when the inertial sensor fails but is not isolated.

**Table 1 sensors-19-02911-t001:** Parameters of SINS.

Error Sources		Error Value
Gyroscope	Constant Drift	0.01°/h
	White Noise Error	0.01°/h
	First-order Markov Drift Error	0.01°/h
	First-order Markov Correlation Time	3600 s
Accelerometer	First-order Markov Bias Error	0.0001 × G s/m^2^
	First-order Markov Correlation Time	1800 s

**Table 2 sensors-19-02911-t002:** Parameters of Sub-system.

System Type	Error Sources	Variance of Error Mean
GPS	Position Error	10 m, 10 m, 20 m (Longitude, Latitude, Altitude)
	Velocity Error	0.1 m/s, 0.1 m/s, 0.2 m/s (east, north, up)
Optical Flow	Velocity Error	0.5 m/s, 0.5 m/s (east, north)
Lidar	Horizontal position error	0.1 m, 0.1 m (Longitude, Latitude)

**Table 3 sensors-19-02911-t003:** The Comparative Data Analysis of Position Errors of the Optical Flow Sensor with and without Faults.

	Longitude Error (m)		Latitude Error (m)		Altitude Error (m)	
	Mean	Mean Square Error	Mean	Mean Square Error	Mean	Mean Square Error
Fault-free	1.02 × 10^−4^	4.42 × 10^−5^	−1.53 × 10^−4^	2.08 × 10^−5^	1.67	6.36
Measurement Fails (Optical Flow)	4.90 × 10^−5^	4.53 × 10^−5^	−7.42 × 10^−5^	4.60 × 10^−5^	−5.23	7.15

**Table 4 sensors-19-02911-t004:** The Comparative Data Analysis of Position Errors of the Inertial Sensor with and without Faults.

	Longitude Error (m)		Latitude Error (m)		Altitude Error (m)	
	Mean	Mean Square Error	Mean	Mean Square Error	Mean	Mean Square Error
Fault-free	1.02 × 10^−4^	4.42 × 10^−5^	−1.53 × 10^−4^	2.08 × 10^−5^	1.67	6.36
SINS with Fault	−1.22 × 10^−4^	5.09 × 10^−5^	−1.58 × 10^−4^	7.37 × 10^−5^	−2.25	8.21

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
