# Peer review of "Research on Adaptive Multi-Source Information Fault-Tolerant Navigation Method Based on No-Reference System Diagnosis"

_sensors, 2019, doi:10.3390/s19132911_

Round 1
Reviewer 1 Report
The proposed self-regulation filter fusion is very practical for heterogeneous multi-source navigation information. Minor revisions are required for the manuscript: p.1 Abstract: last sentence line 27th - ... coefficient is studied --> ... coefficient are studied. p1. Introduction : First sentence line 32nd ... MAV --> MAV(Micro Air Vehicles) p.3 line 98, k --> use mathematical symbol 'k' p.3 For Eq.(3), Eq(4) All the mathematical symbols, X, H, Phi, Z, should be denoted after their Eqs. p.14 line 356 results ---> results. *add stop '.'Author Response
Comments and suggestions are all responsed and revised in line 27,32,98,110,111,358. The details are shown in the word uploaded, highlighted in red.

Reviewer 2 Report
In my opinion this paper can be published after minor improvement of English.
Author Response
1、I'll check about the paper carefully and improve the english.
Response:In my opinion this paper can be published after minor improvement of English.
Reviewer 3 Report
Summary
The present manuscript addresses the problem of fault detection based on no-reference systems. Authors claim some improvements in the existing residual chi-square detection method, and propose an interactive and distributed residual fault detection method. The weight distribution of the reference system and the information distribution coefficient is studied, and a self-regulation filter fusion method based on no-reference fault detection is designed.
Broad Comment
The abstract is difficult to read, being the sentences too long and without appropriate punctuation. The first sentence can serve as an example. This problem, associated with several typos, grammar, and vocabulary issues, can be extended to the all document.
I could not understand the claims that are being under research, and the methods that will be applied to solve them.
The introduction presents several statements without any reference. When references start to appear, they are not explained, neither integrated in the present work.
Section 2 is present without references or claims, thus, it cannot be understand if it’s a contribution from the authors or simple state of the art.
Section 3 continues with state of the art dissemination, but without any reference. It seems that the proposed method starts its explanation above line 184 and continues in section 3.3.
Equations (38) and (39) that should address the novelty of the presented method are not clear and not properly explained.
Simulations are presented without the full information necessary to replicate the results, but above all, there is no comparison with any other state of the art method. Without being compared with other methods, I do not consider the validation of the proposed methodology.
The last sentence of the conclusion “the information distribution coefficient of the filter is adjusted in real time to realize the intelligent and autonomous fusion of multi-source navigation information and improve the reliability and stability of the navigation system under fault” was not discussed, neither proven.
Specific Comments
1. Sentences too long and difficult to read, thus the reader cannot follow the authors' reasoning;
2. Do not use references in the abstract;
3. Claims are not clear;
4. Several statements without any reference;
5. Line 63 with several references not explained or framed in the present work;
6. Line 69, 186, 252, spacing;
7. Table 2, the GPS error is very high, what hardware was considered to apply those values?
(Please consider the annotated PDF)

Author Response
Sentences too long and difficult to read, thus the reader cannot follow the authors' reasoning;
1、Sentences have been divided.
Do not use references in the abstract;
2、References have been deleted in the abstract.
Several statements without any reference;
3、References have been added.
Line 63 with several references not explained or framed in the present work;
4、References in line 63 are basis and the method in this paper is proposed according to them.
Line 69, 186, 252, spacing
5、They are revised in line 69,186 and 252.
Table 2, the GPS error is very high, what hardware was considered to apply those values?
6、GPS signal is selected as a reference, and it is RTK with high precision.
7、Comments and suggestions are all responsed and revised .The details are shown in the document uploaded, highlighted in red.
It has no value without a comparison with other state of the art methods.Cannot validate results.
8、From figure 6 and 7,it is shown that fault can be diagnosed by reference-free fault detection method.While by normal reference detection method, it can not be diagnosed if reference system is with fault.
The last sentence of the conclusion Was not discussed neither proven.
9、The conclusions are based on simulation results, and it is shown that navigation precision is very poor if the reference system is with fault and isn't detected. So, the reference-free detection method can improve the reliability and stability.
The abstract is difficult to read, being the sentences too long and without appropriate punctuation. The first sentence can serve as an example. This problem, associated with several typos, grammar, and vocabulary issues, can be extended to the all document.
10、The long sentences, grammar and vocabulary issuse are revised according to reviewer’s comments.
I could not understand the claims that are being under research, and the methods that will be applied to solve them.
11、Normally, some system is selected as reference system. It is used to judge whether other systems are failure or not. However, what if the reference system is failure? The reference-free detection method is proposed and resolve this problem.
The introduction presents several statements without any reference. When references start to appear, they are not explained, neither integrated in the present work.
12、Rerences are added when the introduction presents several statements, and integrated in the present work.
Section 2 is present without references or claims, thus, it cannot be understand if it’s a contribution from the authors or simple state of the art.
13、References are added to claim the basis of fault model, according to reviewer’s comments.
Section 3 continues with state of the art dissemination, but without any reference. It seems that the proposed method starts its explanation above line 184 and continues in section 3.3.
14 References are added to continue with state of the art dissemination. The references are the same as those in section 2.
Equations (38) and (39) that should address the novelty of the presented method are not clear and not properly explained.
15、Equations(38) and (39) originate from federated filtering theory, and the reference is added.
Simulations are presented without the full information necessary to replicate the results, but above all, there is no comparison with any other state of the art method. Without being compared with other methods, I do not consider the validation of the proposed methodology.
16、Simulations are presented to show how about the navigatioin precision if the reference system is failure,but it is not detected. Comparatively, it is not a problem with the reference-free detection method.
The last sentence of the conclusion “the information distribution coefficient of the filter is adjusted in real time to realize the intelligent and autonomous fusion of multi-source navigation information and improve the reliability and stability of the navigation system under fault” was not discussed, neither proven.
17、As responsed in point 16, the proposed method is used for each system, without any reference system, so it avoid the failure of any reference system, because there is not any reference system. In this case, it can improve the reliability and stability compared with the normal method. The normal method is with a reference system.

Round 2
Reviewer 3 Report
I consider that the necessary clarifications have been given and the changes are in accordance with the suggestions made.